# Parental Influence in Disengagement during Robot-Assisted Activities: A Case Study of a Parent and Child with Autism Spectrum Disorder

SunKyoung Kim [1,*] , Masakazu Hirokawa [2] , Atsushi Funahashi [3] and Kenji Suzuki [2]

1   Faculty of Library, Information and Media Science, University of Tsukuba, Tsukuba 305-8550, Japan
2   Faculty of Engineering, Information and Systems, University of Tsukuba, Tsukuba 305-8573, Japan;
    hirokawa_m@ieee.org (M.H.); kenji@ieee.org (K.S.)
3   Faculty of Sport Science, Nippon Sport Science University, Yokohama 227-0033, Japan; funahashi@nittai.ac.jp
*   Correspondence: sunkyoungk@acm.org

**Abstract:** We examined the influence of a parent on robot-assisted activities for a child with Autism Spectrum Disorder. We observed the interactions between a robot and the child wearing a wearable device during free play sessions. The child participated in four sessions with the parent and interacted willingly with the robot, therapist, and parent. The parent intervened when the child did not interact with the robot, considered "disengagement with the robot". The number and method of intervention were decided solely by the parent. This study adopted video recording for behavioral observations and specifically observed the situations before the disengagement with the robot, the child's behaviors during disengagement, and the parent's intervention. The results showed that mostly the child abruptly discontinued the interactions with the robot without being stimulated by the surrounding environment. The second most common reason was being distracted by various devices in the play sessions, such as the wearable device, a video camera, and a laptop. Once he was disengaged with the robot, he primarily exhibited inappropriate and repetitive behaviors accentuating the symptoms of autism spectrum disorder. The child could re-initiate the interaction with the robot with an 80% chance through the parent's intervention. This suggests that engagement with a robot may differ depending on the parent's participation. Moreover, we must consider types of parental feedback to re-initiate engagement with a robot to benefit from the therapy adequately. In addition, environmental distractions must be considered, especially when using multiple devices for therapy.

**Keywords:** robot-assisted autism therapy; autism spectrum disorder; disengagement; parent intervention; social referencing; NAO robot; wearable device

## 1. Introduction

Various robots have been actively used in therapy for children with Autism Spectrum Disorder (ASD) [1–4]. Since ASD characteristics include deficits in social communication and interaction [5], robots have been used to improve the social communication skills or behaviors of children with ASD in several ways [6,7]. The primary role of robots for children with ASD was as a model of specific behaviors [8], including gestures [9] and facial expressions [10]. In previous studies, humanoid robots, such as Robota and Kaspar, were controlled to move its head, arms, or facial expressions [4]. Children with ASD who participated in the studies watched and imitated the robot's behaviors. One study has even shown that children with ASD tend to interact more with robots than with humans [11].

Another main purpose of using robots was to evaluate the severity of autistic traits [12], provide feedback [13], or elicit specific behaviors [14]. In this case, the role of robots is to assist therapists. Previous studies have reported that using robots positively influenced therapy for children with ASD. In particular, using a robot easily drew the attention of children with ASD from the beginning of the session to the tenth therapy session in

a study [15]. Therapists have also reported that using robots is beneficial for children with ASD in therapies [16]. That is, robots have played a central or supportive role in autism therapy.

However, the role of parents in robot-assisted therapy has not been highlighted. When considering that parents are often included directly or indirectly in therapies for children with ASD [17], it is necessary to examine whether a parent influences a child's interactions with a robot. Although parental interventions for children with ASD have been considered effective in traditional autism therapies [18], studies exploring parents' influence in robot-assisted therapy were rarely conducted. A study investigated the effects of parents' presence on their children's social or behavioral changes, such as engagement with a robot, smiles, and stereotyped behaviors in robot-assisted therapy [19]. The presence of parents influenced their children's behaviors towards the robot positively throughout ten sessions. In particular, children with severe impairments showed higher engagement with the robot when their parents are with them.

Different from the previous study, we examined the influence of a parent's active involvement on a child's engagement in robot-assisted activities. It is a prerequisite to maintain a child's engagement with a robot to facilitate a specific behavior using a robot. Therefore, we focused on free play sessions which can be widely included in play-based autism therapy using a robot, and explored a parent's influence on the moments of a child's disengagement, which was defined as discontinuation of interactions with a robot. In particular, we explored the events of disengagement and parental influences on reinitiating the engagement. We observed what is happening before and during disengagement, and how the parent influences the child's behaviors. We adopted a wearable device as a possible distractor which can cause disengagement. In addition, we observed whether unusual situations during disengagement happened or not and how the parent intervened the moments because a parent can be more influential in unexpected situations as a model of social reference.

We adopted a case study method to explore what kinds of factors should be considered for further research on parental influences during robot-assisted therapy. We video-recorded and observed a child and a parent's case during robot-assisted activities. We explored what is happening between engagement and disengagement and summarized the events quantitatively referring to a reported method [20]. Additionally, we closely observed the moments of disengagement during unusual situations and explored the parental influence qualitatively.

## 2. Background

### 2.1. The Characteristics of ASD

According to the Diagnostic and Statistical Manual of Mental Disorders, ASD is a neurodevelopmental disorder [5]. Neuropsychiatric conditions involve limited and repetitive behaviors inappropriate in social situations. These behaviors are inconsistently and gradually observed in children with ASD. Stereotypic behaviors have a broad scope, including verbal, non-verbal, or motor-oriented cues. Additionally, they can be simple or complex [21]. Stereotypic behaviors can be observed in various body parts such as face, head, arm, leg, hand, fingers, and so forth. Common stereotypic behaviors include shaking head and hands, jumping, running, and spinning [22]. A study investigating social engagement with human peers and stereotypic behavior of three children with ASD showed that simultaneous and motor stereotypic behaviors reduced with increased social engagement with human peers [23]. Therefore, we assessed whether engagement with a robot influenced an increase or decrease in stereotypic behaviors.

### 2.2. Wearable Technology for Children with ASD

Robots and various technologies have been applied to therapy or education for children with ASD [24]. These days, wearable technologies have been highlighted, as they can recognize individual physiological changes in children with ASD and provide real-time

feedback. Physiological parameters helpful for therapy in children with ASD include respiration rate, electrodermal activity, body temperature, cortisol level, blood pressure, electromyogram (EMG), and an electroencephalogram. Various devices have been used to record these parameters. The types of devices used were wristbands, armbands, watches, chest straps, shirts, and glass [24]. Notably, the wristband and glass types showed high suitability for individuals with ASD. They have shown promising results for use in daily life [25]. However, it is still unknown what effects a wearable device will have on children's behavior when that device is considered distracting. In this study, we assumed that a wearable device can be a distractor. Therefore, we considered the EMG data unusable if the participant is distracted by the device, although we recorded EMG using the wearable device throughout the therapy sessions. We focused on the negative influence of the wearable device on the child's engagement with a robot.

### 2.3. Engagement

Maintaining engagement is considered essential for creating long-term interactions and achieving goals between humans and robots [26–28]. Engagement has been investigated and defined using various definitions in human–robot interaction from behavioral, affective, or cognitive perspectives [29,30]. In previous studies, a variety of methods have been applied for measuring different aspects of engagement, including measuring behaviors and physiological changes [24,27,31]. In particular, behavioral cues, such as gestures, head position, gaze, and facial expressions have been the main targets for recognizing humans' engagement [32,33]. To analyze behavioral engagement, recording and analyzing videos have been commonly used in human–robot interaction [15]. In this study, we focused on specific behavioral and affective engagement related to play as we explored a child's engagement during play-based activities. We recorded videos of the free play sessions and observed the play behaviors and smiles of the child toward a robot. We defined engagement as the continuation of play activities or interactions with the robot, including holding hands of the robot and showing gestures to the robot. In contrast, we defined disengagement as the discontinuation of the play activities or interactions with the robot, such as being distracted by surroundings and turning away from the robot.

### 2.4. Social Referencing

Typically developing children learn appropriate behaviors by modeling other people, which is called "social learning" [34]. According to social referencing theories, the primary source of learning is parents' behaviors or responses in an unfamiliar context. Social referencing can be observed in infants even at six months of age [35]. This involves a sequence of unclear or new events wherein a child does not know how to respond. Since the context must be unusual or novel for a child, studies related to social referencing were conducted to arouse feelings, such as fear, pain, or joy. Typically developing children observed others' facial expressions when deciding how to behave in such irregular situations. In contrast, children with ASD in previous studies showed limited to no social referencing [36,37]. On the other hand, a study which investigated social referencing with a robot and a parent from 20 children aged 4 to 5 years found that all participants showed social referencing behaviors with their parents. However, the study setting was to teach novel words, not in an unexpected situation [38]. In this study, we particularly explored parental influence during disengagement moments. When disengagement happened in an unusual situation during play with a robot, we observed the detailed interactions between the child and parent, and examined whether social referencing occurred or not.

### 3. Materials and Methods

#### 3.1. Participants

We selected a 9-year-old boy with a moderate level of ASD from the Institute for Developmental Research of the Aichi Human Service Center in Japan. The child with ASD participated in four play sessions with a robot. This research was approved by the Ethical

Committee of the Aichi Human Service Center following the Declaration of Helsinki and the ethical rules of the institute. The activities and observations were conducted at the same institute. The child's parent provided written informed consent and participated in all sessions.

### 3.2. Robot

An NAO robot (Softbank Robotics, Tokyo, Japan) was used to explore the potential of robot play sessions for individuals with ASD. It is a small-sized (58 cm in height) humanoid robot which has been used for children with ASD facing difficulties in human interactions [8,39]. Particularly, the robot can create physical interactions by moving 25 joints with 25 degrees of freedom [40]. In this study, it was used for physical activities during play therapy for a child with ASD. The robot's behavior was controlled by the Wizard of OZ method [41], which involves remotely controlling a robot using a human operator. It makes users believe they are interacting with an autonomous system while the system is being controlled by a human. In this study, a human operator observed the child's responses to the NAO robot in the observation room and controlled the robot in real time. The operator adjusted the robot's behaviors to draw the child's attention to the robot or maintain the attention. A software called Choregraphe (Softbank robotics, Tokyo) was used to monitor and control the robot's behavior. To ensure safety and separate the play area from the camera recording area, the area in which the child and his parent interacted with the robot was surrounded by a fence (Figure 1).

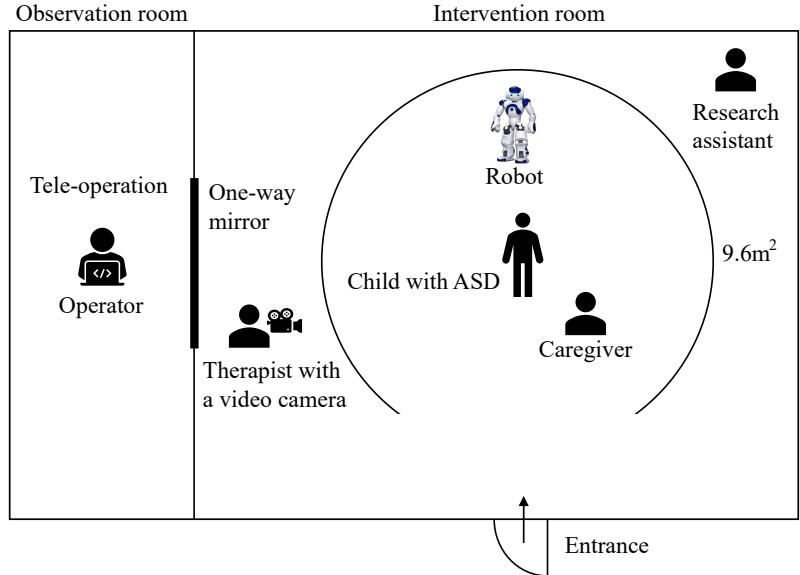

**Figure 1.** Therapy room setting.

### 3.3. Wearable Device

A wearable device was used to record the surface EMG of the child's facial muscles [42]. The device was non-intrusively attached to both sides of the child's face (Figure 2). Facial muscle activities were measured with EMG sensors and monitored wirelessly with a laptop connected to the sensors. Wearable devices with facial EMG sensors have been used to capture facial expressions and measure emotional engagement [24,25,42]. However, we focused on the disengagement with the robot which was caused by wearing the device in this study. Therefore, the EMG data analysis was not included in this study. There were three sizes of the device, including small, medium, and large, and the sizes were adjustable. A small device was attached to the child's face, and the device was adjusted to the smallest size. The child immediately accepted the device and did not show any discomfort during the play session.

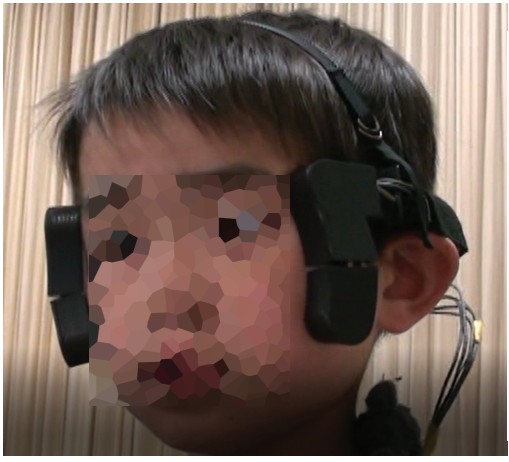 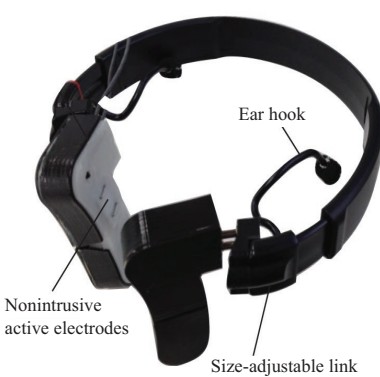

**Figure 2.** The participant wearing the wearable device.

*3.4. Procedure*

The child participated in the total of four sessions with his mother. His mother was asked to help him play with the robot when he does not know what to do with the robot or when he does not focus on the robot. Each session consisted of the preparation, start of the therapy, playtime with the robot, and end of the session. Each session lasted approximately 20 min.

### 3.4.1. Preparation

First, the child and his parent were introduced to a preparation room. The researchers explained the play sessions and introduced a wearable device to the child. The wearable device was attached to the child and tested by the researchers to confirm its proper functioning. Subsequently, the therapist started recording videos, and showed the child 20 visual stimuli that appeared on a laptop screen. The purpose of showing visual stimuli was to calm down the participant and reduce the effects of previous activities, such as running to the therapy place. Each stimulus was presented every two seconds, and all stimuli were considered neutral and did not arouse specific facial expressions. The stimuli were pictures of everyday objects or plants. Additionally, the researchers confirmed whether the child accepted the wearable device during this preparation procedure.

### 3.4.2. Start of the Therapy

After the preparation, the child and his parent were introduced to an intervention room. When the intervention room and participants were ready, the therapist opened the door of the room. Opening the door was a signal for the robot operator and for the research assistant to start the play session. When the intervention room door was open, the operator exhibited greeting behaviors from the robot, such as standing up, turning the head to look around, and waving hands.

### 3.4.3. Play

The task of the child was to interact with the robot for the maximum duration during the robot-assisted activities. The child and his parent participated in a session every two weeks. The therapy schedule was decided by a therapist while considering that building bonds between a child and a robot will require time including a certain interval. In addition, participants' available time was considered.

The child was allowed to interact with the robot, his mother, therapist, and research assistant without restrictions during all sessions. The child freely touched the robot, talked to the robot, and showed gestures.

The robot's behaviors and reactions were improvised by a human operator. It included an imitation of the child's behaviors, and movements of fingers, arms, or head. However,

the robot's voice function was not used intentionally to urge the parent's active participation. As the robot did not ask any specific behaviors to the child verbally, the parent mediated the play between the child and the robot.

The child's mother was asked to participate in all sessions to guide the child to interact with the robot. She intervened in the interactions between the child and the robot without restraints. She also freely decided when, where, and how to intervene in the interactions. For example, she judged situations by watching the robot's movements and the reactions of the child. She suggested an appropriate interaction or play with the robot to the child.

The therapist observed the participants' natural interactions with the robot and recorded their behaviors using a handheld camera. However, the therapist provided guidance only when the child was hesitant to play with the robot. When the child was silently looking at the robot or not doing anything with the robot, the therapist suggested possible activities with the robot to the child, including a rock-paper-scissors game, give-and-receive small balls, or walking with the robot. Depending on the activities suggested by the therapist, the NAO robot was controlled to move fingers, arms, or legs differently.

The research assistant did not interact directly with either the robot or the child. The research assistant entered the fence and fixed the devices to continue the play sessions only when the robot or wearable device failed to function correctly.

### 3.4.4. End of the Session

After 20 min had elapsed, the therapist suggested ending the session. The robot operator created behaviors that indicate a farewell by discontinuing the activities with the child and waving the hand of the robot. After completing a session, the child and his parent returned to the waiting room and detached from the wearable device. Subsequently, the video recording ended.

### 3.5. Video Annotation

We recorded videos of the play sessions and annotated the child's behaviors. Video analysis is a typical method for capturing and measuring human behaviors, particularly in children with developmental disorders, as their behavior shows developmental delays or atypical characteristics [33]. Two trained examiners annotated the behaviors of the child and parent.

In this study, we focused on changes in the child's interactions with the robot. We quantitatively annotated the length and the number of times the child interacted with the robot, parent, or therapist. We also annotated the duration and the number of distractions. The duration of each annotated behavior was measured per millisecond using Dartfish, a software for analyzing videos (Dartfish, Fribourg, Switzerland). From these annotations, we explored how the child disengaged in playing with the robot. The detailed annotation observations are as follows:

### 3.5.1. Child's Interactions

We annotated the start and end times when the child showed verbal or non-verbal behaviors towards the robot, parent, or therapist. When the child was simply looking at the parent, therapist, or robot, we did not consider the behavior as being engaged with them. The child's behavior was annotated as engaged only when he behaviorally or verbally interacted with one of them.

### 3.5.2. Child's Disengagement with the Robot

In this study, the task assigned to the child was to play with the robot. We focused on when the child did not interact with the robot. We defined the moment of disengagement as when the child was distracted by his inner states or by surrounding environmental changes. For instance, the child was considered disengaged when he turned away from the robot. We annotated the start and end times of the child's disengagement with the robot. We also annotated why and which behaviors occurred when the child was distracted. In

particular, we considered repetitive and stereotypic behaviors, a criterion for diagnosing ASD [5], as expected behaviors when the child does not engage in activities with the robot. We observed if the child exhibited repetitive verbal or motor stereotypic behaviors.

### 3.5.3. Parent's Interventions

We annotated the start and end times when the parent provided intervention while the child was distracted. We also observed how the parent intervened and if the child responded to the intervention.

### 3.5.4. Smiles

We annotated the start and end times of the child's smiles to explore how the play sessions with the robot aroused smiles. Smiles were defined as facial expressions characterized by an upward corner of the mouth or a downward corner of the eyes [14].

## 4. Results

We analyzed the annotations of the recorded videos regarding three aspects: (a) the child's interactions with the robot, parent, or therapist, (b) the child's behaviors when he was distracted, and (c) the parent's interventions. The reliability of the two video annotators was high for the annotations of the three aspects. Intraclass correlation coefficient estimates and their 95% confident intervals were calculated using SPSS statistical package version 28 (SPSS Inc., Chicago, IL, USA) based on a mean-rating ($k = 2$), consistency, and two-way mixed-effects model. The average intraclass correlation coefficient was 0.834, with a interval from 0.751 to 0.858 ($F(627, 627) = 6.01$, $p < 0.001$).

### 4.1. Quantitative Description

This study focused on the child's behaviors and his parent's interventions when he disengaged from interactions with the robot. First, we quantitatively counted the durations and numbers of (1) the child's interactions with the robot, (2) the child's interactions with the therapist, (3) the child's interactions with the parent, and (4) the child's disengagement with the robot based on the video annotation.

Table 1 shows the total duration and number of the child's interactions with the robot, therapist, or parent and disengagement with the robot in each session. Moreover, it shows the total duration and the number of the child's smiles in each session. We found that the total duration of interaction with the robot typically increased, although it decreased marginally during the third session. Notably, the total duration of interaction with the robot was the highest during the last session. The number of interactions with the robot showed the same trend as the total duration.

**Table 1.** Duration (seconds) and numbers during the five types of moments.

| Moments | Session 1 | | Session 2 | | Session 3 | | Session 4 | |
|---|---|---|---|---|---|---|---|---|
| | Duration | Number | Duration | Number | Duration | Number | Duration | Number |
| With robot | 376 | 25 | 770 | 30 | 658 | 27 | 1366 | 34 |
| With therapist | 48 | 6 | 118 | 13 | 83 | 6 | 14 | 3 |
| With caregiver | 249 | 18 | 222 | 17 | 69 | 3 | 109 | 10 |
| Disengagement | 699 | 24 | 370 | 8 | 702 | 14 | 458 | 15 |
| Smiles | 653 | 71 | 829 | 90 | 351 | 17 | 561 | 42 |

In contrast, the total duration and number of interactions with the therapist or parent were lower than those of the robot throughout the sessions. Moreover, the duration and number of interactions with the therapist were lower than those with the parent during the first, second, and fourth sessions. This suggests that the child played with the robot rather than interacting with people around him. When interacting with people, he interacted with his parent rather than the therapist. This indicates that the child showed a clear preference

for interacting with the parent over the therapist. Therefore, the benefits of robot-assisted interventions may increase by assigning a role to a parent of children with ASD.

Regarding the interaction with the robot, we observed a trend and possible relationship with the changes in disengagement and smiles throughout the four sessions (Figure 3, Table 1). First, the disengagement duration showed an opposite trend to that of interactions with the robot. This result is expectable because disengagement was defined as turning away from the robot. Another noticeable point in Table 1 is that his interaction time with the therapist or the parent was less than the disengagement time. Then, what was he doing when he was disengaged with the robot? We cover this in the next section.

Second, the duration of smiles showed the same trend as the interactions with the robot. This might be since the engagement with the robot was a fun activity that aroused smiles. However, smiles cannot explain the changes in interaction with the robot completely. The child was engaged with the robot the most during the fourth session while he smiled the most during the second session. The child may have lost interest in the robot over time but was engaged in activities with the robot. Therefore, the motivation to play with the robot can be either internal or external, such as by the parent's directions.

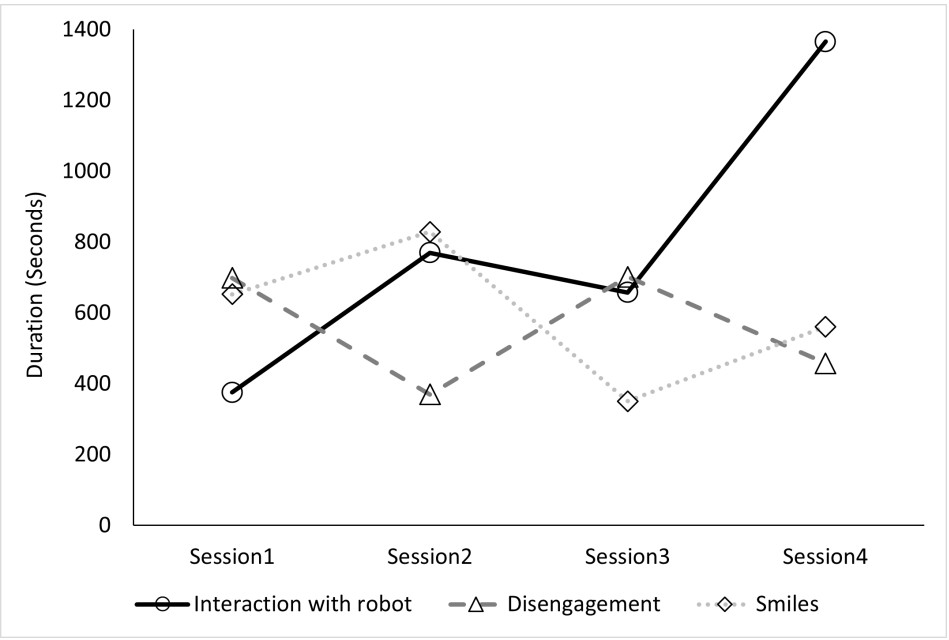

**Figure 3.** Changes in the duration of the interaction and disengagement with the robot and the number of smiles throughout the four sessions.

### 4.2. Detailed Observation of Disengagement

The quantitative description indicates that the child's interaction with the robot increased throughout the four sessions. However, the results raised questions about whether his engagement with the robot was motivated voluntarily and whether he could re-initiate the engagement by himself after disengagement. Therefore, we thoroughly observed the behaviors of the child and parent when they were not engaged in robot interaction. Figure 4 shows a summary of the ratio of disengaged moments. We observed disengaged moments from three perspectives. The first is the child's focus before disengagement with the robot. The second is the behavior that occurred during disengagement. The third is whether intervention by the parent occurred. The disengaged moments throughout the four sessions were observed 61 times (24 times in the first session, 8 in the second session, 14 in the third session, and 15 in the fourth session).

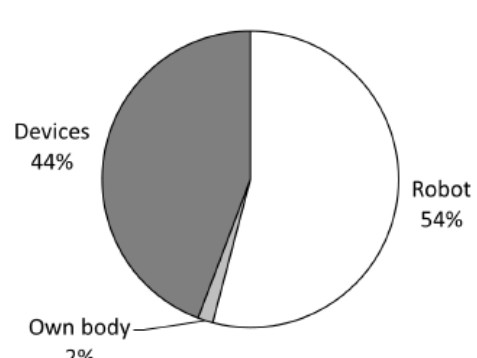

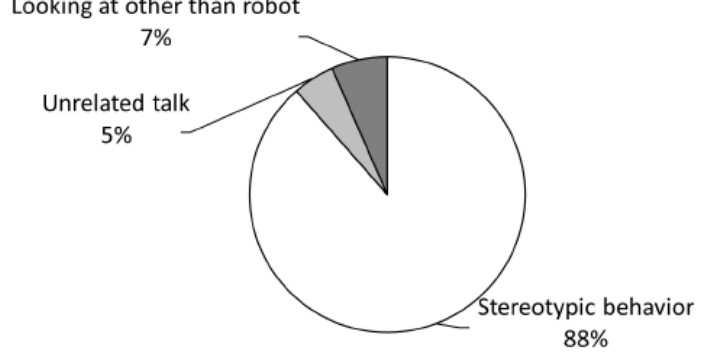

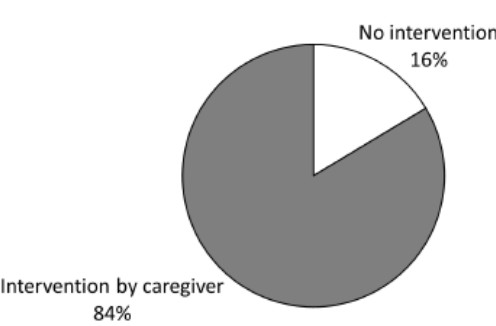

**Figure 4.** The ratio of the child's behaviors and the parent's intervention during the disengagement (total 61 times).

4.2.1. Situation before Disengagement

The most frequent situation that preceded the 61 instances of disengagement with the robot was that the child was interacting with the robot. In these cases, the child abruptly discontinued the interaction with the robot, even though the robot was controlled to show the child various movements, such as standing up, sitting down, showing gestures, or throwing a ball to retain his attention. This may imply that the activities or interactions with the robot were insufficient to maintain the child's engagement.

The second occurrence of most disengagements was when the wearable device or other devices in the therapy room distracted the child from the robot. 52% of the instances of the child being distracted by devices were due to the wearable device, and 45% involved the child being distracted by the therapist's camera. When the wearable device was not fixed well on his face, the child was distracted and disengaged from the robot. Although the device was adjustable and adjusted well before starting the play sessions, his movements, including running and jumping, disturbed the device adjustment from the correct position on his face. Moreover, the child often touched and tried detaching the wearable device during the play sessions, although he accepted the device during preparation time. His touch made the wearable device unstable on his face, leading his attention to the device rather than the robot, thereby, leading to disengagement with the robot. Another device that distracted the child was a laptop monitoring the EMG recordings. This case occurred just once, but it suggests that various devices used for the play sessions might distract children and cause them to have trouble focusing on the robot.

4.2.2. Behaviors during Disengagement

The annotation of the disengaged moments was based on the definition of engagement and disengagement in this study (Table 2). In particular, we observed stereotypic behaviors in the child, unrelated talks, and looking at objects other than the robot during the moments of disengagement. The difference between unrelated talks and stereotypic verbal behaviors in this study was repetition and the length of utterance. When he mentioned repetitive and inappropriate words, we counted them as stereotypic behaviors. In contrast, we counted the behaviour as 'unrelated talks' when he uttered a series of sentences discussing a topic that was unrelated to the situation.

**Table 2.** Annotated events of engagement and disengagement.

| Event | Engagement | Disengagement |
|---|---|---|
| Definition | continuation of the interactions with the robot | discontinuation of the interactions with the robot |
| Example | holding hands of the robot, showing gestures to the robot | turning away from the robot, gaping at someone |

Of the observed behaviors during the disengagement, 88% (54 out of the 61 instances of disengagement from the robot) were due to stereotypic verbal or motor behaviors. After he was distracted by the environment or discontinued his interactions with the robot, he repeatedly showed the same behavioral patterns. Additionally, he repeatedly said the exact words or words of the same category. Various stereotypic behaviors occurred concurrently. A common pattern was standing up suddenly and walking or running around. This pattern was observed 33 times out of the 54 stereotypic behaviors. In addition to the behaviors of abrupt standing up and walking or running around, the child suddenly sang a song eight times or shouted twice. The central verbal stereotypic behavior was related to food and transportation. He suddenly mentioned food names seven times or mentioned train names twice. The most severe stereotypic behaviors were suddenly standing up, jumping, and running around while talking about specific words. It was observed twice right after he discontinued the interactions with the robot in both cases.

A total of 7% of instances of disengagement were the child looking at a person or objects other than the robot. This included the therapist, the therapist's camera, the wearable device, and his own legs. The remaining 5% were the child performing 'unrelated talks'—speaking about something unrelated to his current situation. He abruptly talked about a person who was not present in the room, the upcoming weather, and the room temperature.

4.2.3. Parent's Intervention

When the child was not engaged with the robot, 84% (51 out of the 61 instances of disengagement from the robot) was intervened by his parent. Various types of interventions were observed simultaneously or consecutively by the parent. The parent's intervention included showing him how to interact with the robot and taking the child near the robot physically. The types of parent intervention in order of frequency are as follows (The number in parentheses indicates frequency): talking about the robot to the child (15), holding or blocking the child when he tried to escape from the play area (10), calling the child (9), suggesting an activity to the child with the robot (9), pointing at the robot (7), showing gestures (5), complementing the child (4), taking the child to the robot (4), complementing the robot (2), saying "Look at the robot" to the child (2), saying "No" to the child (2), stroking the robot's head (2), touching the robot (2), showing a mark "X" with crossed arms or fingers to the child (1), and saying "Help the robot" to the child when the robot fell (1).

After the intervention, 80% (41 out of the 51 instances of the parent's intervention) was returning to the robot. When he did not return to the robot immediately after receiving the

parent's intervention, the parent's next intervention was a combination of talking about the robot to the child, calling the child, complementing the child, and talking to the robot.

A noticeable pattern from the child was that he did not look at the parent's face while the parent provided an intervention. However, we observed that he returned to the robot and imitated the parent's behaviors or sayings in 9 out of 41 cases. The frequency of imitated behaviors was as follows (The number in parentheses indicates frequency.): touching the robot (2), stroking the robot's head (1), and singing (1). The child also repeated what the parent had said to them to the robot. The frequency of these repeated phrases was: "You are good" (1), "Are you okay?" (1), "Help the robot" (1), "Let's play again" (1), and "Thank you" (1).

We observed that he imitated or repeated his mother, but he did not directly look at the parent's face. Avoiding direct gaze during social interactions is one of the autistic characteristics [5]. Therefore, it is noticeable that the vocal, verbal, or gestural feedback was imitated by the child. It may suggest that parents can guide a child with ASD to engage with the robot by providing more effective feedback.

*4.3. Qualitative Description in Unexpected Situations*

We observed that more than half of the disengagement happened while the child was interacting with the robot. Therefore, we closely observed the moments of disengagement and the parent's influence, particularly in unexpected situations with the robot. As social referencing behaviors might occur during unexpected situations, we explored if the behaviors occur from the child with ASD and how the parent influence the behaviors.

During the first session, when the child was holding a hand of the robot and the robot was walking, the robot suddenly fell down. The child looked surprised, and turned away from the robot. His mother quickly supported the falling robot, and said "Sorry" to the robot. The child was looking at the situation while keeping distant. The mother showed a gesture to urge him to come to the robot. The child approached the robot a little, but turned away from the robot with a frown on his face. His mother said "It's okay. Come here." Then, the child smiled and approached the robot and stroked the robot's head. He moved back a little bit and said, "Are you ok?" to the robot.

During the second session, when the child was touching the arm of the robot and the robot was moving arms, the robot suddenly stopped and fell down. His mother said, "Ohhhhh" while looking at the falling robot with an anxious face. The child turned away from the robot and went far away from the robot. The mother made the robot stand up, and showed a gesture to urge him to come to the robot. The child was just looking at the robot while smiling. His mother said, "Stroke the robot's head." Then, the child approached the robot and stroked the robot's head, while saying "Are you ok?" to the robot.

During the third session, when the child was touching the fingers of the robot, the robot tried to stand up, but the robot fell down. The child immediately said, "Are you ok?" to the robot and shook the fallen robot. His mother said, "Save the robot." to the child, and the child made the robot stand up while saying "I will save the robot." The robot fell down again, and his mother said with a serious face, "Are you ok?" to the robot. The child also said, "Are you ok?" to the robot. However, the child suddenly stood up and jumped while smiling and saying "Robot, robot, small child." His mother said "Are you ok?" to the robot again with a serious face. The child approached the robot, and said "Are you ok?" to the robot.

During the fourth session, when the child was holding a hand of the robot and the robot was walking, the robot suddenly fell down. His mother pointed out the fallen robot with an anxious face. The child approached the robot, and said "Are you ok?" to the robot. Then, the robot tried to stand up, but it stopped. The child said "Are you ok?" to the robot immediately, and pretended to cry.

His mother consistently tried to teach him how to behave in unexpected situations. Her facial expressions were either worried or serious, and she never smiled in the situations. She tried to make the child say considerate expressions and help the robot. Although the child

did not refer to the mother's facial expressions and he even smiled and jumped during the third session, his mother's verbal expressions were effective to make him follow directions or imitated the expressions. During the final session, he immediately approached the robot and said "Are you ok?" in the expected situation without his mother's intervention. The consistent verbal feedback by the mother may have encouraged the child to say the same verbal expressions in a similar situation.

## 5. Discussion

We explored parental influences and related factors to be considered, including stereotypic behaviors, distracting devices, unusual situations, and social referencing, from a case for further research on robot-assisted therapy. We examined the parental influence on a child's engagement in robot-assisted activities. In particular, we observed what is happening before and during disengagement. In the child's case, we attempted to identify the events which facilitated the continuation of engagement with the robot and the parent's involvement.

First, we observed a trend of interactions between the child and robot throughout the four sessions. The total duration and number of interactions with the robot showed an overall increase. During the last session, the child showed the most extended interactions with the robot and the shortest with the therapist. Since the prerequisite of robot-assisted therapy is engagement in interactions or activities with the robot, the child showed a favorable response to robot-assisted therapy; the child interacted with the robot rather than with his parent or the therapist. However, the smiles decreased during the third and fourth sessions compared to the first and second sessions. Although smiles are expressions with various inner statuses, when considering smiles as an indicator of positive emotions or feelings, the child may have interacted with the robot involuntarily to perform a task during the third and fourth sessions.

Second, we found 61 disengaged moments and observed the child during these moments to establish what motivated him to re-initiate the interaction with the robot. When we observed what happened or the child's action before disengagement with the robot, we observed that 54% of these disengagements were sudden; the child stopped interacting abruptly with the robot. This suggests that the robot interactions might not be sufficient for engaging the child to continue. In this study, we solely controlled the robot's body and created movements. The robot did not talk or make sounds. Therefore, the child interacted with the robot while only receiving visual and tactile stimuli. The robot's limited actions may have resulted in abrupt discontinuation of the interactions. However, 44% of the distractions were caused by multiple devices, including the wearable device, the therapist's camera and the laptop monitoring the EMG recording of the wearable device. Specifically, he often touched the wearable device or tried to detach it, although he showed no resistance during the preparation of each session. This suggests that we need to consider the acceptance of a wearable device for extended periods. Moreover, the number of devices that will be used and exposed to a child with ASD during therapy should be considered. If the devices are the primary source of distraction, this might negate the positive effects of using them.

Third, we observed the child's behavior during disengagement and intervention by the parent. Of the observed behaviors, 88% were stereotypic verbal or motor behaviors considered inappropriate and repetitive. The most common behavioral patterns were suddenly standing up, walking around, or running around. He also spoke repeatedly of specific food or car names when not engaged with the robot. This suggests that these symptoms of ASD may have been soothed while interacting with the robot. In contrast, the symptoms of ASD were prominent when the child was not engaging with the robot. Another explanation could be that focusing on a target, whether a robot or otherwise, can make the child's symptoms less prominent. In 84% of cases, the parent provided intervention during these moments. The parent provided various types of interventions simultaneously or one by one. The most frequent intervention was talking about the robot,

to the robot or calling the child. However, the types of intervention did not always make the child return to the robot. When the child did not interact with the robot after the verbal intervention, the parent attempted physical interventions, such as holding or blocking the child when he tried to escape from the play area. This shows that engaging the child with the robot is not straightforward. It was necessary to stop violent, repetitive stereotyped behavior and call the child's attention to the robot.

Fourth, the child's chances of interaction with the robot increased by 80% after receiving the parent's intervention. Notably, he also imitated the parent's behaviors or words in 9 of 41 cases. All imitated behaviors were related to hand movements and vocal behaviors. He imitated the parent's behavior or words appropriate in the situation. As with previous research results [36,37], he had limited eye contact with people around him and did not look at the parent's face directly while receiving the parent's feedback during his disengagement. This shows that he could refer to the parent's responses when the feedback involved voice or hand movements, but not facial expressions. The qualitative description particularly shows that the child re-initiated the interactions with the robot by referring to his mother's verbal reactions in an unusual situation where the robot suddenly stopped or fell. Although social referencing in children with ASD has been reported limited, this study indicated that the parent facilitated a specific behavior of the child in unexpected situations and played a role as a social referencing. In addition, it should be noted that the dynamics among the child, the robot and the parent can be intentionally adopted for future robot-assisted therapy to facilitate an appropriate behavior in a social context.

Although this study made some notable observations during robot-assisted activities, this case study has limitations. This research is based on a case of a child with ASD and his mother. The child was diagnosed with a moderate level of symptoms, but the details of his conditions, such as details of comorbid conditions, were not disclosed when we recruited the participant through the Institute for Developmental Research. In addition, the influence of a parent might be different depending on the parent's personality or relationship with the child. However, we did not include their interactions outside of the research settings. Moreover, we focused on the quantitative analysis of their behaviors. Another limitation is that the therapy included only play sessions and that did not target specific behavioral changes. Therefore, the depth of analyzing this case has limitations.

There are also limitations with the wearable device which was used in this study. There is a possibility that the device was not enough comfortable for using it throughout a session, which was around 20 min. The participant might have been bothered by the wearable device made of plastics on his face and head. If the device had a softer material and it was a smaller size covering the face and head minimally, different results might have been obtained. Otherwise, it is also possible that the child had high sensory sensitivities related to symptoms of ASD. It was challenging to apply a wearable device that satisfies an individual's comfortableness. Therefore, it should be noted that this case might be extreme to show the negative influences of the wearable device.

However, this study provides critical insights for future research. First, the effects of parents' intervention should be considered for maintaining engagement with a robot for autism therapy; parental intervention may increase the benefits of robot-assisted therapy. In addition, future research should consider what kinds of feedback from a parent will be more effective. Second, the child was distracted by not only the wearable device but also a video camera and a laptop. This suggests that the use of multiple technologies should be carefully considered to take full advantage of using either robots or wearable devices. Third, social referencing of a child with ASD can be considered during robot-assisted therapy involving parental participation. It is important for children to learn how to behave in an unfamiliar situation. Therefore, the new role of a robot and a parent, such as a robot assisting a parent, needs to be considered to create a therapeutic setting for social referencing.

The limitations of this study should be addressed by future research. Future studies must collect more extensive data from broader populations with ASD. Moreover, the personalities of parents or relationships between a child and a parent need to be considered.

The timing, way, or purpose of parental involvement also should be further investigated. Additionally, future research will cover the effects of parent involvement in robot-assisted therapy for improving a specific social or interpersonal behavior of children with ASD.

## 6. Conclusions

Parents may be a crucial factor to re-initiate engagement in robot-assisted therapy. To obtain the benefits of using a robot for autism therapy, it is essential to maintain the engagement of individual children with ASD with the robot. This study showed that the child returned to the robot with an 80 percent change after receiving intervention from the parent. The most favorable scenario is that the robot is engaging continually, and the child is not distracted by the environment or their own behaviors triggered by neurobiological conditions. However, robot-assisted therapy in a natural setting may lead to unusual or distracting situations. In addition, therapists may consider using various technologies or devices. Therefore, this study suggests that the moments of disengagement should be considered in robot-assisted therapies and how to provide feedback to a child disengaged with a robot including parental involvement.

**Author Contributions:** Conceptualization: S.K.; methodology: S.K., M.H. and A.F.; software: M.H.; validation: S.K., M.H., A.F. and K.S.; investigation: S.K., M.H. and A.F.; data curation: S.K., M.H. and A.F.; writing (original draft preparation): S.K.; writing (review and editing): S.K.; supervision: A.F. and K.S.; project administration: A.F. and K.S.; funding acquisition: S.K. and K.S. All authors have read and agreed to the published version of the manuscript.

**Funding:** This work was supported by JSPS KAKENHI Grant Number JP20J11008 (Grant-in-Aid for JSPS Fellows). In addition, this work was supported by Japan Science and Technology Agency (JST) for the Core Research for the Evolutional Science and Technology (CREST) research project on Social Signaling (JPMJCR19A2).

**Institutional Review Board Statement:** The study was conducted according to the guidelines of the Declaration of Helsinki, and approved by the Ethical Committee established by the Aichi Human Service Center.

**Informed Consent Statement:** Informed consent was obtained from all subjects involved in the study.

**Data Availability Statement:** The video annotation for this study can be found in an online repository at: doi:10.7910/DVN/K0EPIV.

**Conflicts of Interest:** The authors declare no conflict of interest. The funding organization had no role in the design of the study, in the collection, analyses, or interpretation of data, in the writing of the manuscript, or in the decision to publish the results.

## Abbreviations

The following abbreviations are used in this manuscript:

ASD     Autism Spectrum Disorder
EMG     electromyogram

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
