# Peer review of "Parental Influence in Disengagement during Robot-Assisted Activities: A Case Study of a Parent and Child with Autism Spectrum Disorder"

_mti, doi:10.3390/mti6050039_

Round 1

Reviewer 1 Report

Overall, this is an excellent and insightful case study that has been very well presented by the authors. I commend them on their hard work and am happy to recommend this paper for publication. This paper presents a very interesting case study and provides insights into potential future research regarding the role parents and caregivers can play in interventions designed for children with ASD.

I have provided a list of suggested corrections below, the majority of which are correcting typos or improving the wording of certain sentences.

Line 44 – typo – change “showed a limited or lack of social referencing” to “showed limited to no social referencing”

Line 72 – the phrase “we observed if” could read better as “we assessed whether”

Line 81 – please add references for the various types of wearable technologies used in previous research

Line 83 – the sentence starting “However” is a little unclear, perhaps it should be written as “However, it is still unknown what effects a wearable device will have on children’s behaviour when that device is considered distracting.”

Line 85 –  Based on reading the methods section I believe that rather than saying the robot and device were used to measure engagement, it is more accurate to say that the study’s main aim was to examine the child’s engagement with the robot and the role of the parent/caregiver in maintaining the child’s engagement. The intention was to also use a wearable device to measure facial expressions correlated with engagement, however the experimenters discovered that the wearable device was distracting for the child and so rather than use it for its initially intended purpose, they instead focused on its effect as a distractor. I suggest rewriting this sentence to clarify this, perhaps mentioning that the distracting nature of the wearable device meant that the EMG data it recorded was not usable.

Line 91 – the sentence starting “In previous studies” is a bit unclear. It could be rephrase to something like “in previous studies, a variety of methods have been applied for measuring different aspects of engagement”

Section 2.3 – It would be good to have a more concrete definition of the engagement and disengagement behaviours you looked for. Could you please provide an example of what observable behaviours were considered ‘engagement’ vs. ‘disengagement’?

Lines 124 – 128 – the initial purpose of the wearable device was to supplement video analysis of facial expressions. Your statement in the lines I’ve highlighted indicate that this purpose changed but is not clear on how. After reading the rest of the paper I understand that, whilst the child responded well to the device initially, they later found it distracting and would touch it/try to remove it. I imagine this impacted data collection and made the EMG data unusable. It would be helpful if you could clarify this in this section.

Section 3.4 – Overall the procedure is clearly defined here. However, you don’t mention how many sessions there were until the results section. I think it would be good if you add in the number of sessions to this first section. I’m also curious about whether the child was accompanied by the same parent each time and whether the parent was given any special instructions or simply told to try to keep the child engaged. These details aren’t essential but if you have space it would be great to add them in where relevant.

Section 3.4.1 – I'm unclear on the purpose of showing the child the visual stimuli? Was this just to give the child something to do whilst the experimenters assessed how comfortable the device was? Or was it to configure the device in some way?

Line 166 – Reword the sentence to “The research assistant did not interact directly with either the robot or the child”

Line 187 – I’m a little unclear of what is meant by “annotations of the child’s interactions did not include simply watching them”. Does this mean that the child was not annotated as engaging with anyone if they were simply looking at someone (parent, therapist, robot), but that the child’s behaviour would only be annotated if they behaviourally or verbally engaged with someone? Please clarify.

Section 3.5.2 – the overall definition of disengagement here is clear. The only suggestion I’d make is to add an example to the end of the sentence on lines 191-192. For instance, was the child considered disengaged if they turned away from the robot, or did they have to go up to another person or object first before being considered disengaged?

Line 211 – please report the statistical analysis used. E.g. “(One-way ANOVA: F(627, 627) = 6.01, p < .001).”

Line 229 – I agree with the conclusion that the role of the parent is important and should be considered when designing and conducting interventions. However, it might be helpful if this conclusion were clarified a little by giving an example of what considerations could be made. For example, it might be worth stating that because the child shows a clear preference for interacting with the parent over the therapist, interventions might benefit from having the parent play a particular role in the intervention as they may have more success engaging with their child than a therapist.

Lines 230-242 – this section was a little confusing. I suggest rewording this section to make your findings and conclusions clearer. It seems that the results indicate that as the child’s engagement with the robot increased, there was a decrease in the amount of disengagement behaviours observed. However, whilst there seems to be a relationship between disengagement behaviours and the frequency of smiles such that smiling increased as disengagement decreased, there does not seem to be as strong a relationship between smiling and interaction with the robot (Figure 3), at least for session 4. Additionally, you point out that time spent not interacting with the robot was not necessarily spent interacting with the parent of therapist instead (Table 1). However, I’m not sure what is meant when by the statement that “careful observation of the child’s behaviours when not engaged with the robot” is necessary. Does this mean that it would be valuable to study what the child is doing when not engaged with the robot in order to determine what might be distracting them? Please expand.

Line 255 – This first sentence is a little confusing. I’ve provided a suggested rewording: “The most frequent situation that preceded the 61 instances of disengagement with the robot was that the child was interacting with the robot. In these cases, the child would abruptly discontinue the interaction…”

Line 262 – when reporting percentages in text please also state what it is a percentage of. So for example “52% of the instances of the child being distracted by devices were due to the wearable device, and 45% involved the child being distracted by the therapist’s camera.”

Line 271 – I suggest rewording this to something like “might distract children and cause them to have trouble focusing on the robot”

Section 4.2.2. – Reword section title to “Behaviours during disengagement”

Line 277 – I suggest rephrasing this to “we counted the behaviour as ‘unrelated talks’ when he uttered a series of sentences discussing a topic that was unrelated to the situation”

Line 279 – The statement in brackets can be simplified to make it clearer: “(54 out of the 61 instances of disengagement from the robot) were due to stereotypical verbal or motor behaviours.” This same format can be used for lines 299 and 312

Line 291 and 294 – since the authors have already defined what the percentages are, the rest of these statements can be made much shorter. For example “7% of instances of disengagement were the child looking at a person or objects other than the robot.” And “The remaining 5% were the child performing ‘unrelated talks’ – speaking about something unrelated to his current situation.”

Line 296 – typo “He abruptly talked about a person who was not present…”

Line 302 – typo “robot and taking the child near the robot physically.”

Line 304/305 – there shouldn’t be a line break here

Line 306 – the last item on this line should be “suggesting an activity with the robot to the child (9),”

Line 313 – Just to make this sentence flow more smoothly I suggest rewording “the type of parent’s intervention was combinations” to “the parent’s next intervention would be a combination”

Lines 318-323 – No line break. Also, I suggest splitting this into two lists just so it’s clearer and less wordy. Something like: “The frequency of imitated behaviours were as follows: touching the robot (2), stroking the robot’s head (1), singing (1). The child also repeated what the parent had said to them to the robot. These repeated phrases were: “You are good” (1), “Are you okay?” (1), “Help the robot” (1), “Let’s play again” (1), and “Thank you” (1).”

Line 324-325 – The concluding statement is a little confusing. I did note that you made a point in the Discussion that because the child didn’t look at the parent’s face he wasn’t able to imitate facial expressions. I think this applies to your conclusion on line 324-325. Please expand on this sentence just so the conclusions are clear.

Line 329 – rather than saying you attempted to observe the continuation of engagement, would it be more accurate to state that you attempted to identify the events which facilitated the continuation of engagement with the robot? 

Line 345 – typo, “not sufficiently be engaging the child to continue” should be “not be sufficient for engaging the child to continue”

Line 347 – typo, I think “textile” should be “tactile”

Line 350 – typo, “or the laptop” should be “and the laptop”

Line 360 – for the sake of being specific I would suggest changing “the symptoms of ASD” to “these symptoms of ASD”

Line 365 – typo, “talking about the robot, to the robot” (just need to add the comma)

Line 369 – typo, “violently repetitive” suggests that it was the repetition, not the behaviour, that was violent. I believe this sentence should read “It was necessary to stop violent, repetitive stereotyped behaviour and call the child’s attention to the robot.”

Line 382 –  I suggest rewording this to “such as details of comorbid conditions, were not disclosed when we recruited…”

I think you can afford to remove the paragraph where you discuss the study’s limitations. It is true that, because this is a case study, the findings can’t be generalised but personally I don't consider this a limitation because it is the nature of a case study. I suggest instead presenting this as something the reader should note and be aware of, rather than a limitation. For example, “It should be noted that because this is a case study, the findings and conclusions we present cannot be generalised. However, this case report provides critical insights for future research….”

Line 395 – the sentence “However, this may cancel the effects of using the technology.” is unclear to me. I’m not sure what is meant here, please clarify.

Figures and Plots – for plots using multiple colors, please consider using either different shades of the same color (e.g. light grey, dark grey and black) or other textures, markers or patterns to ensure that the plots are more accessible, for example for people who are color-blind

Reviewer 2 Report

This paper presents a case study which examines one child’s experience within four 20-minute robot-assisted play sessions. The presented data includes behavioral analysis of the child and his parent.

The Methods need a specific section elaborating play activities and the procedure. What were the core motivations behind those activities? The main limitation of this study is that the activities do not target specific social skills such as joint attention, emotion labeling, turn-taking etc. It is confusing to refer to this study as having a robot-assisted therapy. I would suggest using robot-assisted play instead.

Due to the ethnographic nature of most case studies, I also expected to read about child characteristics (both medical and individual), preferences, and needs. It is a single case study and therefore can potentially combine both qualitative and quantitative data about the child. It is indeed unfortunate that the paper does not discuss EMG data obtained using wearable device. I was looking forward to learning these findings.  

The authors claim using a case study design, which in fact includes a mix of diverse data collection methods such as qualitative observations. A good amount of HRI studies (mainly conducted by Kerstin Dautenhahn and Ben Robins) with children with ASD have used this method and provided helpful tips for conducting case studies in HRI. It would be good if you referred to these studies for study design inspirations.  

The authors rely on a frequentist approach to data analyses and presentation. Case studies tend to focus on why-questions about what's behind the numbers. I couldn't get that kind of in-depth information from this study.

In Introduction, there are three different formulations for what the authors are exploring. These are very confusing for a reader. Could you please revise these statements and maybe specify research aims/questions more explicitly?

Lines 49-50: whether robot-assisted therapy can be effective long-term without parental influence?

Lines 57-59 : how a parent influences a child’s continued engagement with the robot, specifically in unusual situations with a robot and wearable device?

Lines 59-60: We adopted a case study method to explore the various behaviors and responses observed between a child and parent in an uncontrolled therapy environment.

Discussion may benefit from a more comprehensive interpretation of the main results. I don't think this section responded to your initial research questions sufficiently. What was your take on the results as researchers? Are there any supporting or contrasting data from prior studies/theories/knowledge?

The recent paper on parental involvement is quite relevant. I would suggest referencing it in the introduction:

Amirova, A., Rakhymbayeva, N., Zhanatkyzy, A., & Sandygulova, A. (2022). Effects of Parental Involvement in Robot-Assisted Autism Therapy. Journal of Autism and Developmental Disorders, 1-18. https://doi.org/10.1007/s10803-022-05429-x

Minor comments:

  • 33: "Children with ASD accept robots in a short time and engage in interactions with them [14]". Here the authors refer to the long-term study and yet present it as a short-time interaction. It should be rephrased.
  • Line 40: "Social referencing can be observed in infants even at six months of age." It would be good if you cite a source.
  • Line 60. It is mentioned to be an uncontrolled therapy environment. What do you mean by that? The WoZ was used for the control of robot actions, and the therapist could intervene and suggest what activities to play.
  • Section 2.3: Thank you for explaining how you define engagement during the interventions. This section may benefit from creating a table with criteria for coding engagement and disengagement events before discussing them in 4.2.2.
  • Wearable devices are good to present more objective data although researchers need to understand potential challenges that come with it. As you mentioned in Discussion, it might not be comfortable for young children, especially for those who are sensitive to new things and environments. Please discuss its challenges and limitations of the device that you used.
  • Section 3.4.3. I'd like to know the reason why the researchers had a long gap between sessions: "The child and his parent participated in a session every two weeks." Was that purposeful? ASD interventions are usually expected to be regular and intensive.
